# Decrease of heart rate variability during exercise: An index of cardiorespiratory fitness

**Denis Mongin**[1]*, **Clovis Chabert**[2], **Manuel Gomez Extremera**[3], **Olivier Hue**[4], **Delphine Sophie Courvoisier**[1,5], **Pedro Carpena**[3], **Pedro Angel Bernaola Galvan**[3]

**1** Faculty of Medicine, University of Geneva, Geneva, Switzerland, **2** Institute for Advanced Biosciences (IAB), Grenoble Alpes University, Grenoble, France, **3** Department of Applied Physics II, E.T.S.I. de Telecomunicación, University of Malaga, Malaga, Spain, **4** ACTES laboratory, UPRES-EA 3596 UFR-STAPS, University of the French West Indies, Guadeloupe, France, **5** Quality of Care Unit, University Hospitals of Geneva, Geneva, Switzerland

* denis.mongin@unige.ch

**Data Availability Statement:** All the R code used for the analysis is available at https://gitlab.com/dmongin/scientific_articles/-/tree/main/decrease_HRV_effort. The data used for this study have been

## Abstract

The present study proposes to measure and quantify the heart rate variability (HRV) changes during effort as a function of the heart rate and to test the capacity of the produced indices to predict cardiorespiratory fitness measures. Therefore, the beat-to-beat cardiac time interval series of 18 adolescent athletes (15.2 ± 2.0 years) measured during maximal graded effort test were detrended using a dynamical first-order differential equation model. HRV was then calculated as the standard deviation of the detrended RR intervals (SDRR) within successive windows of one minute. The variation of this measure of HRV during exercise is properly fitted by an exponential decrease of the heart rate: the SDRR is divided by 2 every increase of heart rate of 20 beats/min. The HR increase necessary to divide by 2 the HRV is linearly inversely correlated with the maximum oxygen consumption (r = -0.60, p = 0.006), the maximal aerobic power (r = -0.62, p = 0.006), and, to a lesser extent, to the power at the ventilatory thresholds (r = -0.53, p = 0.02 and r = -0.47, p = 0.05 for the first and second threshold). It indicates that the decrease of the HRV when the heart rate increases is faster among athletes with better fitness. This analysis, based only on cardiac measurements, provides a promising tool for the study of cardiac measurements generated by portable devices.

## Introduction

The human heart is involved in the response to the energy demand of the body [1]. Its regulation is mainly driven by the subtle balance between the sympathetic and parasympathetic branches of the autonomic nervous system [1, 2]. The activity and relative level of these two cardiac neural systems cause the main dynamical changes of heart rate (HR) in response to external stimulus, and on a shorter time scale, the fluctuations of the heart R-wave to R-wave (RR) time interval known as heart rate variability (HRV). HRV during rest has been shown to be influenced by psychological [3] as well as physiological factors such as age, body mass index, diseases [4], heart functions and heart diseases [5, 6], body position [7] and physical

made available at https://doi.org/10.13026/2qs3-kh43.

**Funding:** DM: grant IZSEZ_0183540 from the Swiss National Foundation for Science DSC: project fund 100019_166010 from the Swiss National Foundation for Science The funders had no role in study design, data collection and analysis, decision to publish, or preparation of the manuscript.

**Competing interests:** The authors have declared that no competing interests exist.

fitness [8, 9]. During physical exercise, HRV dynamics is drastically modified due to the break of the balance between both branches of the autonomic nervous system.

The progressive withdrawal of the parasympathetic activity and the subsequent increase of the sympathetic activity causes extensive changes in RR intervals. For example, it is well established that the variability of RR intervals (HRV) decreases both in time and frequency according to several HRV indexes, including coarse graining spectral analysis [10], Poincaré plots [11], HRV spectral power in different frequency bands [12–14], or statistical properties of the increments of RR intervals [15, 16]. But the decrease of HRV is not the only effect of exercise: it is also known that exercise modifies the linear correlations of the RR intervals as measured by Detrended Fluctuation Analysis [17], and produces a reduction of their sample entropy [18], pointing to a reduction of the complexity of the cardiac signal associated to exercise. Similar conclusions can be drawn by measuring the nonlinear correlations of RR intervals, which are also known to decrease with exercise [19, 20].

The decrease of HRV during exercise has been described in most cases, mainly qualitatively, as a function of exercise intensity, measured by the oxygen consumption expressed in percentage of the maximum oxygen consumption (%$VO_2$max) [7, 8, 11, 13]. Lewis and co-authors [12] modelled the decrease of the absolute high frequency (HF) and low frequency (LF) power of the RR series spectrum as an exponential decay of the workload. They find that the decay time (the HRV decay constant) of such exponential regression is correlated to the maximum work capacity of the athletes. The HRV decay constant constitute thus a promising index that could be used as a proxy of a person' cardiorespiratory fitness (CRF). The CRF quantifies the ability to transport oxygen from atmosphere to the mitochondria, and is thus closely related to athletes' performances, but is also a strong predictor of cardiovascular disease and all-cause mortality in the general population [21]. The possibility to use the HRV decay constant to estimate the CRF using only cardiac data stemming from portable heart rate measurement devices, such as Holters or chest heart rate monitors, would be a major benefit. Indeed, CRF is usually measured either directly by the measured peak oxygen consumption, or derived from the maximum work rate achieved during a test, both methods requiring an important material setup. Estimating the CRF from cardiac measurements acquired during variable effort would extend the estimation of CRF to easier to implement effort test, such as the 6 min walk test, and to field measurements.

To do so, the approach of Lewis and co-authors has to be modified so that the HRV decay constant calculation does not rely on workload measurements, which cannot be measured with simple cardiac portable devices and are often not available from field measurements. Furthermore, the use of the workload as the independent variable in the calculation of the HRV decay constant is not valid during recovery periods. Indeed, during these periods, the HRV will increase back to its resting value and will be associated to the same null workload, thus invalidating the exponential relation between the workload and the HRV. We propose instead to analyze the decay of HRV during exercise as a function of the HR, because the latter is regulated during the exercise by the balance of the parasympathetic and sympathetic neuronal activity, which are both at the source of the HRV.

In the present study, we will therefore test the feasibility and the validity to calculate the HRV decay constant using the heart rate instead of the workload as the independent variable when modelling the exponential decay of HRV along exercise. This approach can be applied to resting, exercise and recovery periods, and only requires cardiac measurements as provided by portable devices. To distinguish the HRV from the global change of the RR intervals due to the increase of HR caused by the workload changes, we will estimate the change of RR along the effort test using a recent validated dynamical model [22] to obtain a detrended RR series. We furthermore propose to use the standard deviation of the detrended RR intervals (SDRR) to

estimate the HRV instead of spectral based calculation. Indeed, SDRR has been shown to be closely related to physical performances [23], is considered as a standard measure of HRV [4] and has the advantage of its simplicity, thus being easily implemented and reproduced. Frequency based measurements of HRV on the other hand may require to know the breathing frequency [12], which would not be available when using only cardiac measurements, and depend on the frequency limits used to define the low or high frequency bands of the spectrum [4] or on the norm used to express their values [8], thus leading to sometime contradictory results [24, 25].

We will therefore in the present retrospective analysis first study if the HRV can be modeled as an exponential decay of the heart rate, and compare it with other models using the work load or the work intensity as the independent variable. In a second part, we will calculate the HRV decay constant based on heart rate for each individual and examine how it is linked with classical CRF indices (peak oxygen consumption, maximum work load and power at ventilatory thresholds).

## Methods

### Participants

The database used in this work [26] consists of records of a cycling graded effort test (GET) (Cf. the section below for details) performed by 18 young athletes (10 males and 8 females; 15.2 [14.0, 16.8] year-old, 174.0 cm [165.0, 182.0] height and 62.9 kg [54.3, 76.5] weight, see Table 1) of the Regional Physical and Sports Education Centre (CREPS) of French West Indies (Guadeloupe, France), belonging to a national division of fencing (n = 10), or a regional division of sprint kayak (n = 6) and triathlon (n = 2). All athletes completed a medical screening questionnaire, and a written informed consent from the participants and the legal guardians was obtained prior to the study. The study was approved by the CREPS Committee of Guadeloupe (Ministry of Youth and Sports), the ethics committee of the University of French West Indies and performed according to the Declaration of Helsinki. A short summary of the physiological characteristics of the studied group is presented in Table 1.

### Graded effort test measurement

GET were performed at the end of the off-competition season. The participants performed under the supervision of a doctor in sport medicine a GET on an SRM Indoor Trainer

**Table 1. Physiological characteristic of the 18 participants.** Values indicated are medians [Inter quartile range]. $VO_2$max: maximum oxygen consumption; MAP: Maximum aerobic power; Peak HR: peak heart rate reached during maximal effort test; $PVT_1$: power at the first ventilatory threshold; $PVT_2$ power at the second ventilatory threshold.

| Variable (unit) | Value |
| --- | --- |
| Number | 18 |
| Age (years) | 15.0 [14.0, 16.8] |
| Sex (Male) | 12 (66.7%) |
| Weight (kg) | 62.9 [54.3, 76.5] |
| Height (cm) | 174.0 [165.0, 182.0] |
| $VO_2$max (mL/kg/min) | 36.5 [32.6, 41.8] |
| MAP (W) | 222.5 [177.5, 297.5] |
| Peak HR (beat/min) | 187.2 [183.3, 190.1] |
| PVT1 (W) | 105.5 [79.2, 143.0] |
| PVT2 (W) | 169.0 [141.2, 246.8] |
| sport | Fencing: 10; Kayak: 6; Triathlon: 2 |

electronic cycloergometer (Schoberer Rad Meßtechnik, Jülich, Germany) associated to a Meta-lyzer 3B gas analyzer system (CORTEX Biophysik GmbH, Leipzig, Germany). The room was climatized and did not have external light to provide similar temperature, humidity, and light for each GET. The participants were instructed not to take alcohol, caffeine, nor to practice intense sport activities during the 24 hours preceding their GET. All athletes were boarder of the CREPS and were followed by a nutritionist, thus the diet and eating time were similar all the athletes of the study and the food intake was taken at least 2 hours before the exercise testing. The GET consisted of a 5 min of resting time before exercise, a 3 min cycling period at 50 watts, followed by a workload increase of 15 Watts every minute until exhaustion. Athletes were considered as exhausted when they were not able to maintain a pedaling rate over 60 rotation/min. At the end of the test, measurements were prolonged during a 3 min period to record the physiological recovery of the athletes. The participants were sitting during this recovery period.

Respiratory parameters were recorded breath-to-breath all along the test session. The ventilatory thresholds 1 (VT1) and 2 (VT2) were calculated using the Wasserman method [27]. Maximum aerobic power is the maximum power achieved during the last completed step of the GET. Peak HR and $VO_2$max are the maximum values of the HR and $VO_2$ averaged over 5 breaths. The RR series were derived from Electro Cardiogramm (ECG) recordings (Cardio 110BT, Customed, Ottobrunn, Germany, with 12 derivations). The time resolution of the ECG recording was 1 ms. Heart rate (HR) is calculated as where RR is in millisecond and HR in beat/min.

$$HR = \frac{60000}{RR} \qquad \text{Eq 1}$$

## Analysis

Each individual RR series was first detrended to remove the global RR changes due to the heart rate adaptation to the workload changes. The detrending has been performed by a tested and characterized dynamical model based on simple physiological considerations [12, 21, 22] and free from ad-hoc parameters (see "detrending" subsection). HRV during each individual effort was then quantified by the SDRR, calculated on adjacent windows of one minute (corresponding to each effort step during effort, see "Heart rate variability" subsection). The mean HR, the mean work intensity $\%VO_2$max and the mean work load were calculated in the same windows. The resulting series of SDRR values was then analyzed in two different parts.

In a first part, three models describing the evolution of the HRV during graded effort test (GET) were compared: the first one represents SDRR as an exponential decay of HR, the second one as an exponential decay of the exerted power, and the last one as an exponential decay of the work intensity. More precisely, these models were operationalized as follow:

$$SDRR_j = b_{HR}e^{\frac{-\ln(2) \times HR_j}{\tau_{HR}}} \qquad \text{(model 1)}$$

$$SDRR_j = b_{P}e^{\frac{-\ln(2) \times P_j}{\tau_{P}}} \qquad \text{(model 2)} \qquad \text{Eq 2}$$

$$SDRR_j = b_{I}e^{\frac{-\ln(2) \times I_j}{\tau_{I}}} \qquad \text{(model 3)}$$

where $HR_j$ is the mean $HR$, $P_j$ the mean work load exerted and $I_j$ the mean work intensity calculated in same windows $j$ as the $SDRR_j$. $\tau_{HR}$, $\tau_P$ and $\tau_I$ are the HRV decay constant for these three models (the amount of $HR$, $P$ or $I$ it takes to divide SDRR by 2), and $b_{HR}$, $b_P$ and $b_I$ the

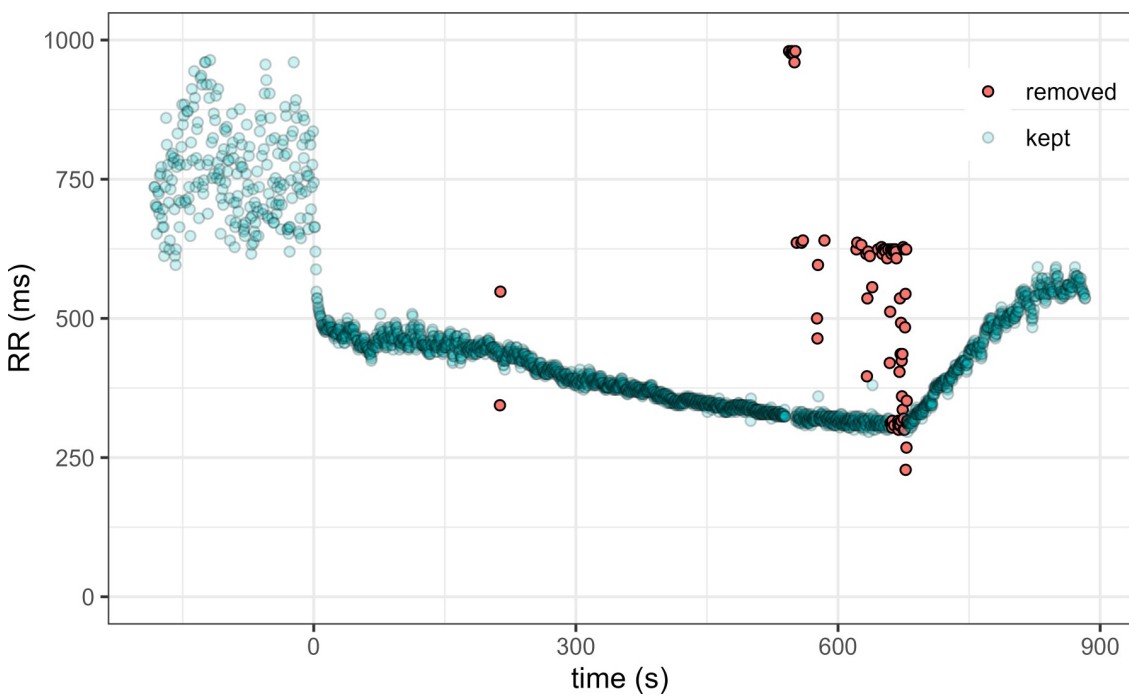

**Fig 1. Example of RR series cleaning.** Measured RR interval during a maximal graded effort test (effort starts at time = 0). The points removed by our cleaning procedure are indicated in red.

HRV intercepts, i.e. the SDRR corresponding to a hypothetical null heart rate, workload or work intensity respectively. The HRV decay constant as defined by Lewis and co-authors [12] is $\tau_P$.

These three models have been implemented by performing least squares nonlinear regressions on the ensemble of the SDRR measurements covering the entire exercise test (measures before the GET and during the recovery are included), and on those during the effort only.

In a second part, we used model 1 on each individual SDRR series and tested whether the estimated decrease of HRV as a function of HR is correlated with CRF. Therefore, we performed the least squares nonlinear regression of model 1 (Eq 2) for each individual SDRR series, and calculated the correlation between the obtained coefficients (the HRV decay constant $\tau_{HR}$ and the HRV intercept $b_{HR}$) and CRF indices (namely $VO_2$max, maximum aerobic power, power at ventilator thresholds) using Pearson linear correlation. The robustness of this analysis has been tested by an extended sensitivity analysis (see "sensitivity analysis" subsection).

## Data cleaning

Prior to performing the statistical analysis, we removed the artifacts in the RR series $\{x_1, x_2, \ldots, x_n\}$ due to connection errors in the electrodes according to the following steps:

- All RR intervals with a value above 1000 ms during effort were removed. This concerned 0.02% of the RR measurements.

- At each point, if the RR value exceeded 2 times or was inferior of the half of the median value of the RR calculated in a 201 RR values windows centered on $x_i$, it was removed. This concerned 0.02% of the measurements.

- At each point $x_i$, if the absolute RR change $x_i - x_{i-1}$ exceeded 10 times the median value of the RR increases calculated in a 201 RR values windows centered on $x_i$, the point was removed. This concerned 0.5% of the measurements.

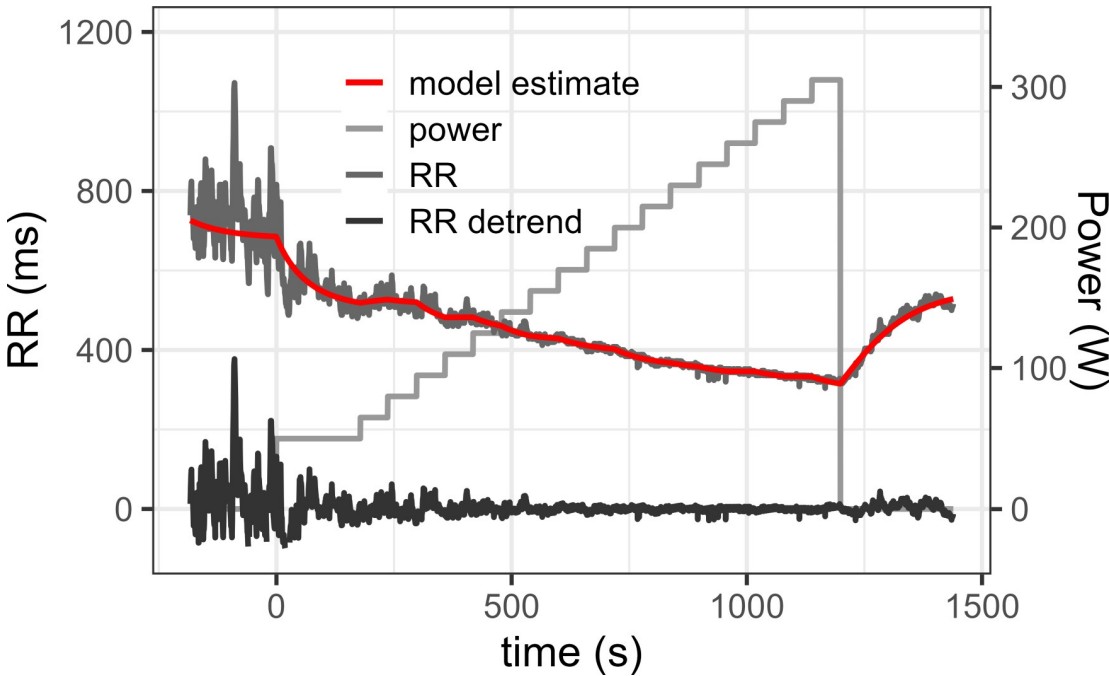

**Fig 2. Example of RR series detrending.** Measured RR interval during a maximal graded effort test, its estimation by a first order differential equation and the resulting detrended RR values obtained by subtracting the model estimate to the real data.

An example of points cleaned in an original raw RR series is presented in Fig 1.

### Detrending

We used recent developments in dynamical analysis to model the non-stationary aspect of HR. The main trend of HR dynamics during a GET can be modeled by a simple first order differential equation driven by the power expenditure. A two-step estimation procedure, consisting in first estimating the time derivative of HR using a spline regression and then obtaining the constant coefficient of the differential equation through a linear regression, produces unbiased estimation of the differential equation parameters [28]. An estimated curve can then be produced by numerical integration of the differential equation with the obtained coefficients. This simple model produces indices sensitive to CRF and performance changes [22]. The possibility to estimate the gain (the amplitude of the HR increase corresponding to a workload increase) for each power step of the exercise test allows to reproduces up to 99% of the HR dynamics during the GET, and yields coefficients varying consistently with the metabolic changes associated to the respiratory thresholds [29].

Because of the absence of ad-hoc parameters and the fact that it has been theoretically and practically validated, this procedure to estimate HR during exercise was used in the present study to detrend the RR time series.

### Heart rate variability

Given a series of $n$ detrended RR intervals $\{x_1, x_2,...,x_n\}$, the series of standard deviations of the RR intervals SDRR calculated on a successive window of $\omega$ RR intervals is

$$sd(\{x_1, x_2, .., x_\omega\}), sd(\{x_{\omega+1}, \ldots, x_{2\omega}\}), \ldots, sd(\{x_{k\omega}, \ldots, x_n\})$$

Where *sd* is the standard deviation, and *k* the integer part of *n*/*ω*. This calculation has the advantage of its simplicity, thus being easily reproducible. We have considered a window size of 1 min for the SDRR calculation in the main study, so that they correspond to each power step during the exercise test.

## Sensitivity analysis

In order to test the sensitivity of our results to the approach proposed, we performed a sensitivity analysis by:

- testing more classical polynomial detrending methods of different orders [30] to obtain stationary RR time series, instead of our parameter-free approach based on HR dynamical model;

- varying the windows size *ω* used for the SDRR calculation.

The polynomial detrending procedure consist in subtracting to each point of the RR series the estimated polynomial trend estimated within a centered windows of *ω* points.

In more detail, it can be described as follows: let us consider a time series of RR intervals and let be *ω* (odd integer) the window size and *p* the polynomial order. For each value of the RR series {$x_i$}, we perform a least squares polynomial regression of order *p* on the data inside the window of size *ω* centered at position *i*. The detrended value at position *i* ($x_{det,i}$) is then obtained by subtracting the estimated value produced by the polynomial regression $\hat{p}_i$ to the experimental value $x_i$:

$$x_{det,i} = x_i - \hat{p}_i$$

During the sensitivity study, the following parameters have been varied:

- The degree *p* of the polynomial has been set to *p* = 0, 1 and 2 (respectively equivalent to substract the mean, the linear fit and the quadratic fit inside the window).

- The window size *ω* (size of the windows for the polynomial detrending, and the size of the adjacent windows used to calculate SDRR) has been set to odd integers between 5 and 101.

We thus tested the robustness of our analysis for 49x3 = 147 different evaluations of SDRR change during effort for each of the 18 participant's RR measurements.

## Statistical analysis

All signal processing and statistical analysis have been performed with the R 4.1 open source software [31]. The comparison between regression models is based on Akaike information criterion (AIC) and Bayesian Information criterion (BIC) [32], which are two estimator of the prediction error used to compare regression models. The calculation of the correlations between the estimated HRV decay coefficients and the CRF indices has been performed using Pearson linear correlation coefficients *r* [33]. The regression of the three models proposed in equation is operationalized with a nonlinear least square regression using the packages *nls*. The *ggplot2* was used for graphical representation [34] and *data. table* for data manipulation. All code used to perform the analysis and generate the tables and figures can be found at the following Gitlab repository: https://gitlab.com/dmongin/scientific_articles/-/tree/main/decrease_HRV_effort.

## Results

The median number of RR distance recorded during the GETs was 2726 beats (Inter Quartile Range [2433; 3548]). The dynamical model based on first order differential equation was used

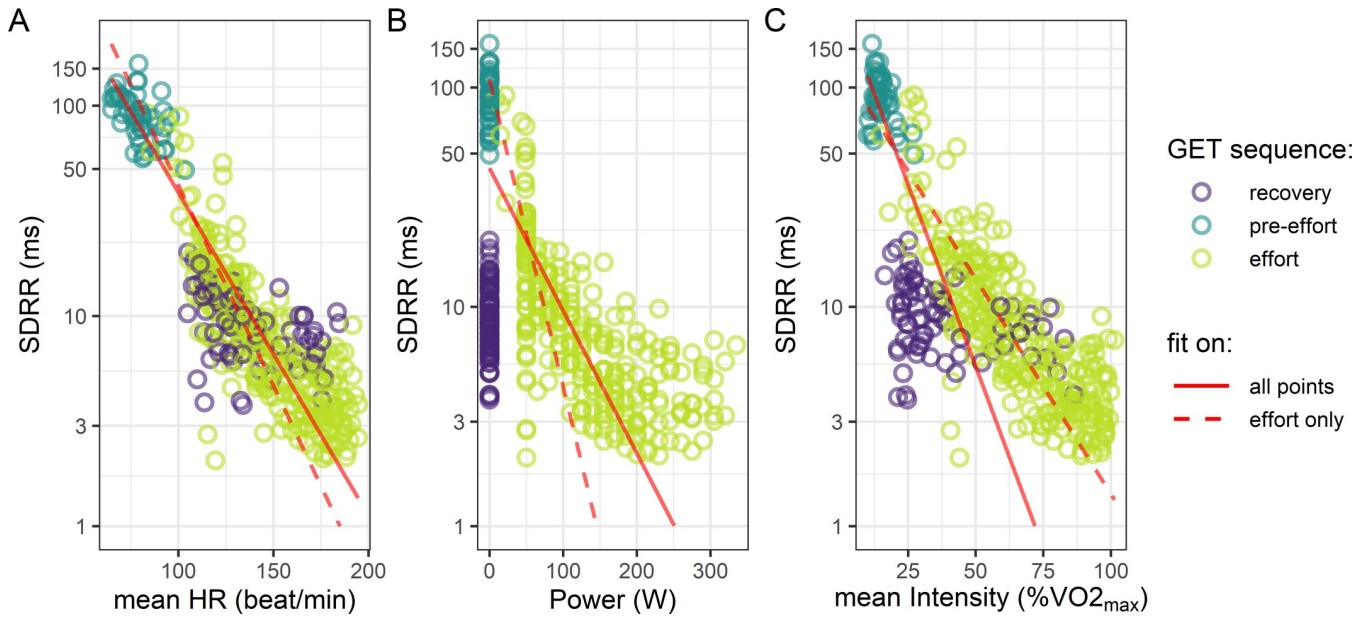

**Fig 3. Different representations of HRV during effort.** SDRR of the detrended RR time series recorded during a graded effort test as a function of: A) the mean heart rate (HR) during the interval, B) the work load during the exercise test and C) the mean work intensity. The color of the points indicates the SDRR was calculating on RR before (pre-effort), during (effort), or after (recovery) the effort. The solid red line corresponds to the regression estimate obtained on the entire set of SDRR values using model 1, 2 and 3 of Eq 2 in A, B, and C, respectively, and dashed red lines are the regression estimate obtained on SDRR values during effort only (i.e. excluding pre-effort and recovery periods) using model 1, 2 and 3 of Eq 2 in A, B, and C, respectively.

to estimate the global changes of RR along each GET, and yielded individual estimations of RR with a median R2 of 0.97 [IQR: 0.91–0.98]. An example of detrended RR estimated using this dynamical model is illustrated in Fig 2. The calculation of SDRR, mean HR, mean power or mean work intensity in adjacent windows of 1 minutes for each detrended RR series yielded individual series of measurement with a median number of 22 [19, 26] points.

## Comparison of models

In this first part we want to determine which representation of the evolution of SDRR along exercise corresponds best to an exponential decay. The ensemble of the SDRR values along the GET of all 18 participants are represented in Fig 3 on a logarithmic scale as a function of the corresponding mean HR (Fig 3A), as a function of the corresponding work load exerted during the exercise test (Fig 3B) or as a function of the corresponding work intensity (Fig 3C). When plotted as a function of HR (model 1, Fig 3A), the ensemble of SDRR measured before, during and after the exercise align nicely on a linear-log scale, indicating a clear exponential behavior. When displayed as a function of the mechanical workload (Fig 3B, model 2) as proposed by Lewis and co-authors [12], the values of HRV at high work rate do not align with the rest of them in a linear-log scale. Furthermore, the HRV calculated before and after the effort have a wide variety of values for the same null power. Finally, when represented as a function of %VO$_2$max, although SDRR values plotted in linear-log scale display a clear linear trend during exercise, they do not align with the values calculated before and after effort (Fig 3C model 3).

When performing the least square nonlinear regressions of these three models on the entire dataset (the estimated model is plotted as a solid red line in Fig 3), the model 1 in Eq 2 describing SDRR as an exponential decay of the mean HR has a significantly lower AIC and BIC than

**Table 2. Akaike information criterion (AIC) and Bayesian Information criterion (BIC) for the models 1, 2 and 3 in Eq 2 when applied to the ensemble of the SDRR computed on different part of the exercise test: Before test (pre), during test (effort) or during recovery (post).**

| model | effort range | AIC | BIC |
|---|---|---|---|
| model1 | pre + effort + post | 3179 | 3191 |
| model2 | pre + effort + post | 3855 | 3867 |
| model3 | pre + effort + post | 3477 | 3489 |
| model1 | effort | 1976 | 1987 |
| model2 | effort | 2080 | 2091 |
| model3 | effort | 2068 | 2079 |

the two others (see Table 2). This results holds when considering cardiac measurement only during, i.e excluding HRV before and after exercise (the estimated model is plotted as a dashed red line in Fig 3). For these reasons, we will use this model for the rest of our analyses.

The mean coefficients of model 1 in Eq 2 estimated on the ensemble of the detrended SDRR series are $b = 1325$ ms and $\tau_{HR} = 19.6$ beats/min ($p < 0.0001$ for both coefficients), meaning that for young athletes, an increase of around 20 beats/min of HR divided SDRR by 2.

## Study of individual decrease of HRV during effort with model 1

Each individual SDRR series had a median of 22 measures [IQR 11, 13, 20–26]. The correlations between the individual parameters $\tau_{HR}$ (the HRV decay constant) and $b_{HR}$ (the HRV intercept), obtained when performing the nonlinear regression of model 1 on each detrended RR series, and the aerobic performances indexes, are reported in Table 3.

The HRV decay constant $\tau_{HR}$ of SDRR as a function of HR is strongly and significantly inversely correlated with the maximum power reached during effort test MAP (r = -0.62, p = 0.006) and with VO₂max (r = -0.60, p = 0.009), and moderately inversely correlated with the power at the ventilatory thresholds (r = - 0.53, p = 0.02 and -0.47, p = 0.05 for PVT1 and PVT2). The fact that the HRV constant, i.e. the HR increase it takes to divide SDRR by two, decreases when the CRF indices increases means that the decrease of SDRR with HR is faster among athletes with better CRF.

Performing the same analysis but separately on males and females yielded higher linear correlation coefficients, but only significant for the correlation between $\tau_{HR}$ and VO₂max ($\rho = -0.68$, p = 0.01) and MAP ($\rho = -0.66$, p = 0.02) for males.

The HRV intercept $b_{HR}$ does not present a significant linear correlation with any of the CRF indices, although it yields a significant positive correlation using rank Pearson

**Table 3. Pearson linear correlation coefficients r with the associated p value between the estimated HRV decay coefficients obtained with model 1 (HRV decay constant $\tau_{HR}$ and the HRV at HR = 0 $b_{HR}$) and cardiorespiratory fitness indices: Maximum oxygen consumption (VO2max), maximum aerobic power (MAP), maximum experimental heart rate Peak HR power at the first and second ventilatory threshold (PVT1 and PVT2) and heart rate recovery (HRR).**

| | $\tau_{HR}$ | | | $b_{HR}$ |
|---|---|---|---|---|
| | Pearson correlation *r* | P value | Pearson correlation *r* | P value |
| VO₂max | -0.60 | 0.009 | 0.38 | 0.12 |
| MAP | -0.62 | 0.006 | 0.40 | 0.1 |
| PVT₁ | -0.53 | 0.02 | 0.30 | 0.22 |
| PVT₂ | -0.47 | 0.05 | 0.22 | 0.38 |
| Peak HR | -0.03 | 0.91 | 0.20 | 0.42 |
| HRR | 0.51 | 0.03 | -0.42 | 0.08 |

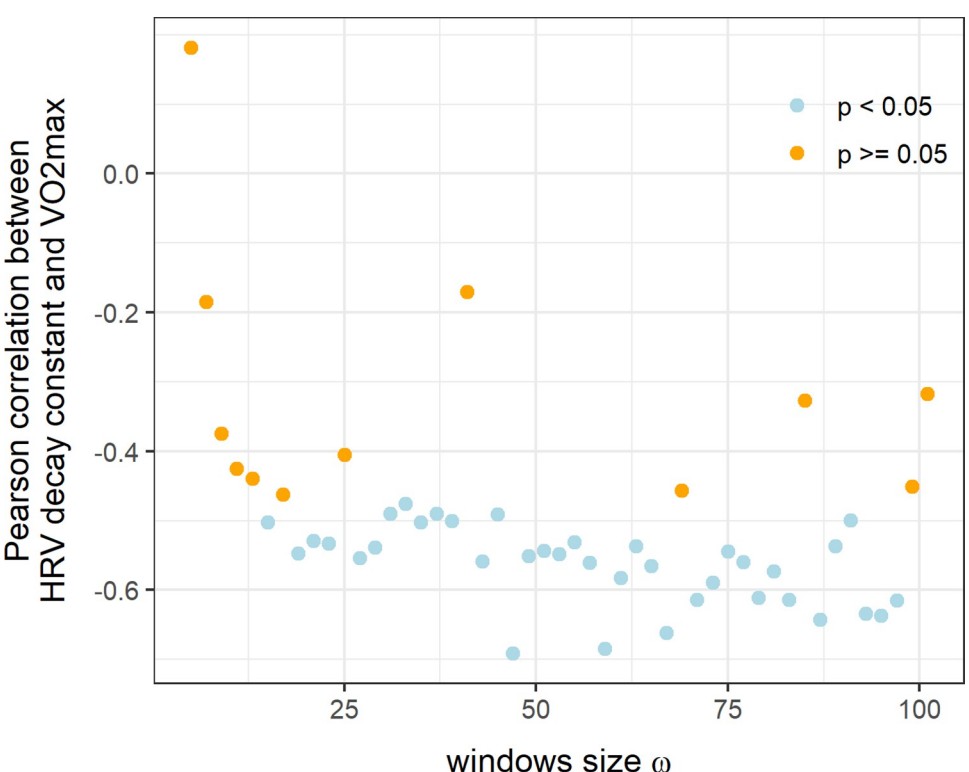

**Fig 4. Results of the sensitivity analysis.** Pearson correlation coefficient and associated p value ($p < 0.05$ or $p >= 0.05$) found between HRV decay constant and $VO_2$max as a function of the windows size $\omega$ when using a $0^{th}$ ($p = 0$) order polynomial detrending.

correlations ($\rho = 0.6$, $p = 0.009$ with $VO_2$max, $\rho = 0.6$ and $p = 0.004$ with MAP, $\rho = 0.64$, $p = 0.005$ and $0.64$, $p = 0.004$ respectively for PVT1 and PVT2). This indicates that participants with better CRF tend to have a higher SDRR at low HR.

## Sensitivity analysis

In Fig 4, the correlation (and the associated p value) between HRV decay constant $\tau_{HR}$ and $VO_2$max are represented for all tested $\omega$ values when using $0^{th}$ order local polynomial detrending. The sensitivity analysis demonstrates that the previous results can be obtained using a simpler $0^{th}$ order local detrending with $\omega > 50$ (i.e. performing the local polynomial detrending in windows of at least 50 points and calculating SDRR in windows of at least 50 points). The results of the sensitivity analysis for other polynomial order and other CRF indices (ventilatory thresholds and maximum power) are presented in S1 Fig.

## Discussion

Our study aimed at validating the HRV decay constant approach developed by Lewis and co-authors using only cardiac measurements and a simple characterization of the HRV. The HRV quantified by the SDRR varies as an exponential decay of HR during effort and recovery, and decreases by 2 every increase of 20 beats/min of HR. This fast decay causes the SDRR to reach values close to the minimum allowed by the resolution of the ECG device early during the GET, thus explaining the absence of significant differences between HRV measured during constant load exercise at high intensities [15]. On the other hand, the exponential decay

constant of SDRR extracted by a nonlinear regression during an incremental exercise is correlated to several parameters linked with CRF, such as $VO_2max$, MAP, and power at the ventilatory thresholds. The linear correlation of 0.6 between the maximum aerobic power and the HRV decay constant we estimate is consistent with the results reported by Lewis and co-authors [12]. This HRV decay constant decreases (i.e. a faster decay of HRV when HR increases) for athletes with higher cardiorespiratory fitness. The analytical approach proposed only requires cardiac measurements and make use of measurements before, during and after the exercise.

This study is to our knowledge the first one proposing to observe the change of HRV during exercise as a function of the corresponding heart rate. When compared to the approach studying the change of HRV as a function of the workload, our approach provide a better exponential variation of HRV and allows to consider period of measurements with a null workload and a strongly varying HRV, such as recovery periods. Compared to the more common approach consisting in studying the change of HRV as a function of exercise intensity, our representation is better described by the exponential decay during exercise and unifies under the same model the HRV calculated on cardiac measurement before and after the effort. The similarity of these two representations during the effort resides in the linear relation that HR and $VO_2$ have when performing graded exercise tests [35]. Their differences reside in the shorter dynamical time of $VO_2$ compared to HR, and is revealed at exercise onset and exercise cessation [22, 36].

Previous studies reported higher HRV for trained participants than untrained participants at rest or at low exercise intensity [37, 38]. This result finds its roots in the higher vagal (parasympathetic) neuronal activity for trained participants compared to untrained ones, as proved by the increase of vagal-related indices of resting and post-exercise HRV [39]. Post-exercise HR recovery studies have also shown that trained participants have a faster re-activation of their vagal activity at exercise cessation [40]. The rate at which HRV decreases when the HR increases during activity is another measure of vagal activity, which has been shown to be directly linked with the exercise capacity [41], explaining thus the link between rate of HRV change during exercise and CRF. In our study, individuals with higher CRF start their exercise with a higher HRV due to their high vagal tone and have a faster subsequent decrease of HRV when their HR increases during exercise due to the faster withdrawal of their parasympathetic activity.

## Strengths and limitations

The correlation between our HRV decay characterization and performance indices found among a heterogeneous population of athletes in term of sport modality is a strength. Indeed, although these different sport modalities require unequal sources of energy and train distinct physical capacities, leading to different cardiorespiratory and cardio autonomic control features, the exponential decay of HRV as a function of HR seems to be a robust indicator of the cardiovascular fitness. On the other hand, the limited number of athletes in each sport category did not allow us to compare the changes of HRV during exercise between sport modalities. The use of a physiologically motivated model of HR dynamics during exercise using no ad-hoc parameters to obtain the detrended RR series and the extensive sensitivity study associated shows the robustness of our approach and facilitates future similar studies by providing guidelines of necessary data acquisition and analysis methods used.

Nevertheless, the limited age range and physiological characteristics of the participants are limitations, and further studies are needed to generalize our results to a more diverse population. The cross-sectional aspect of our approach should be also complemented by a

longitudinal approach. The rather moderate correlation found between HRV decay constant and CRF indices is not high enough to use our analysis to predict CRF at the individual level. Nevertheless, this correlation is similar to those obtained between heart rate recovery and VO₂max or maximal workload [42–44]. Therefore, the HRV decay rate could find applications similar to those of heart rate recovery, such as the use of threshold values to detect cardiovascular problems or risk of deaths [45–47], the monitoring of CRF or training changes [39], or being part of more complex equation with higher predictability [48, 49].

## Conclusion

The present work demonstrates that the measure of the SDRR decay during exercise offers a solid index of cardiorespiratory fitness. Our study proposes a simple model to describe the changes of HRV with effort: SDRR behaves as an exponential of the heart rate. The characteristics of this exponential decay of HRV are highly dependent on the physical capacities and on the cardiorespiratory fitness. The proposed analysis, relying only on cardiac measurements and based on a set of simple mathematical tools, pave the way to the measurement of cardiorespiratory fitness using measurements provided by mobile devices.

## Supporting information

**S1 Fig. Entire sensitivity study.** Pearson correlation coefficient and associated p value ($p < 0.05$ or $p > = 0.05$) found between HRV decay constant and VO2max, maximum aerobic power (MAP), power at the first (PVT1) and the second (PVT2) ventilator threshold as a function of the windows size $\omega$ when using a $0^{th}$ (p = 0), first (p = 1) or second (p = 2) order polynomial detrending.
(TIFF)

**S1 File.**
(DOCX)

## Author Contributions

**Conceptualization:** Denis Mongin, Pedro Carpena, Pedro Angel Bernaola Galvan.

**Data curation:** Denis Mongin.

**Formal analysis:** Denis Mongin.

**Funding acquisition:** Denis Mongin.

**Investigation:** Denis Mongin, Clovis Chabert.

**Methodology:** Denis Mongin, Delphine Sophie Courvoisier.

**Resources:** Clovis Chabert, Olivier Hue.

**Supervision:** Delphine Sophie Courvoisier, Pedro Carpena, Pedro Angel Bernaola Galvan.

**Validation:** Denis Mongin, Manuel Gomez Extremera, Delphine Sophie Courvoisier, Pedro Carpena, Pedro Angel Bernaola Galvan.

**Visualization:** Denis Mongin.

**Writing – original draft:** Denis Mongin.

**Writing – review & editing:** Denis Mongin, Clovis Chabert, Manuel Gomez Extremera, Olivier Hue, Delphine Sophie Courvoisier, Pedro Carpena, Pedro Angel Bernaola Galvan.

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
