## [Decision Letter · Decision Letter 0]

24 May 2022

PONE-D-22-11838Decrease of heart rate variability during exercise: an index of cardiorespiratory fitnessPLOS ONE

Dear Dr. Mongin,

Thank you for submitting your manuscript to PLOS ONE. After careful consideration, we feel that it has merit but does not fully meet PLOS ONE’s publication criteria as it currently stands. Therefore, we invite you to submit a revised version of the manuscript that addresses the points raised during the review process. Among important limitations, rewritting large part of the manuscript is mandatory to specify novelty of the study and clarify Methodological points, including clinical evaluations and data analyses. Please submit your revised manuscript by Jul 02 2022 11:59PM. If you will need more time than this to complete your revisions, please reply to this message or contact the journal office at plosone@plos.org. Please include the following items when submitting your revised manuscript:A rebuttal letter that responds to each point raised by the academic editor and reviewer(s). You should upload this letter as a separate file labeled 'Response to Reviewers'.A marked-up copy of your manuscript that highlights changes made to the original version. You should upload this as a separate file labeled 'Revised Manuscript with Track Changes'.An unmarked version of your revised paper without tracked changes. You should upload this as a separate file labeled 'Manuscript'.

We look forward to receiving your revised manuscript.

Kind regards,

Laurent Mourot

Section Editor

PLOS ONE

Journal Requirements:

When submitting your revision, we need you to address these additional requirements.1.

“DM: grant IZSEZ_0183540 from the Swiss National Foundation for Science

DSC: project fund 100019_166010 from the Swiss National Foundation for Science

“DM: grant IZSEZ_0183540 from the Swiss National Foundation for Science

DSC: project fund 100019_166010 from the Swiss National Foundation for Science

Reviewers' comments:

Reviewer's Responses to Questions

**Comments to the Author**

1. Is the manuscript technically sound, and do the data support the conclusions?

Reviewer #1: Partly

Reviewer #2: Partly

Reviewer #3: Yes

2. Has the statistical analysis been performed appropriately and rigorously? 

Reviewer #1: Yes

Reviewer #2: Yes

Reviewer #3: Yes

3. Have the authors made all data underlying the findings in their manuscript fully available?

Reviewer #1: Yes

Reviewer #2: Yes

Reviewer #3: Yes

4. Is the manuscript presented in an intelligible fashion and written in standard English?

Reviewer #1: Yes

Reviewer #2: Yes

Reviewer #3: Yes

5. Review Comments to the Author

Reviewer #1: In this study, the novelty of the study seems to be unclear. The authors should describe the novelty of this study a little more clearly. In addition, compared to previous studies that evaluated VT and LT, the method used in the previous study (using exercise intensity as the independent variable) seems to have better results than using heart rate as the independent variable in the present study from the perspective of evaluating exercise capacity. Therefore, it is unclear why the authors used the research method with heart rate as the independent variable in the present study.

Furthermore, it is unclear why the authors used the measure of SDRR in this study, even though it is LF and HF in the frequency analysis in the previous study (citation #11). The authors should describe the reasons.

The following three items are posted in the Introduction. The authors should provide details on these three items in the Discussion.

"i) the inherent need to remove the main RR decrease due to the metabolic response to energy expenditure during exercise to obtain a detrended RR series, that is the beat-to-beat variability commonly designed as HRV; ii) the technicity and the variety of the HRV characterizations, as i) the technicity and the variety of the HRV characterizations, as illustrated by the variety of spectral measurements used and by the sometimes contradictory results they provide (20,21); iii) the use of the mechanical work rate during exercise as the independent variable work rate during exercise as the independent variable of the exponential model, which could restrict the use of such approach only on RR series recorded during effort (HRV before and after). during effort (HRV before and after effort are associated to the same power value: 0 Watt) and may be protocol dependent."

Methods; P 6/18; line 2

There is no mention of a rule regarding diet on the day of the study. The authors should describe any dietary stipulations, such as eating at least hours before exercise. Also, should provide detailed information on the exercise test, such as resting time before exercise and number of pedaling rate.

RESULTS;

It is unclear why the authors needed to include RECOVERY in the analysis. If you include it in the analysis, you should provide an explanation for this.

Reviewer #2: The present study „Decrease of heart rate variability during exercise: an index of cardiorespiratory fitness“ focuses on the effects exercise on HRV behavior as a potential marker of cardiorespiratory fitness.

I thank the authors for their effort and submission of the manuscript. All in all, the topic is very interesting and innovative having a non invasive method for HRV assessment and its practical application.

Unfortunately, there are no line numbers in the manuscript.

Abstract:

- Please use HRV as abbreviation after introduction

- There are no results reported in the abstracts, can you provide any numbers and statistic outcomes?

Key Words:

- Some letters are capitalized

Introduction:

- First sentence, reference is missing.

Methods:

- Please provide participant information at first, than study design, than anaylsis.

- First section, which effort is meant, which type of exercise

- Please provide reference for: Akaike information criterion (AIC) and Bayesian Information criterion (BIC).

- Please use VO2max consistently as abbrev.

- Do you use any ranges/threshold values for statistics analysis, eg Spearman coefficient

Participants:

- First sentence: one bracket to much

- In Table 1 you report very low VO2max values for young athletes, please check values

GET:

- Please use the word participants not subjects (in the whole manuscript)

- Page 6: Participants were sitting during…

- …averaged other 5 breaths? „over“?

Heart rate var.

- …in order them?

- …both being processes? (I recommend correction reading by a native speaker)

Statistical analysis

- Some repitition to earlier provided information, please rewrite

Results

- Correlation interpretation in methods section is missing, r values of 0.5/0.6 is at a moderate level.

Discussion

- First section: „strongly related“ there are only corr. coeff. values of around 0.5/0.6; please revise

- „This HRV decay rate increases (i.e. a faster decay of HRV when HR increases) for athletes with higher aerobic capacity.“ Where do you get this result, there is no subsection for different performance level.

- „The strong correlation between our HRV decay characterization and performance indices found among a heterogeneous population of athletes in term of sport modality is a strength.“ Please revise strong correlation (see above)

- All in all, what do the results mean for the individual athlete in ist application (last sentence of conclusion: „ave the way to the measurement of cardiorespiratory fitness using measurements provided by mobile

- Devices“)? On group level you have moderate correlations with performance markers, that given, I would say its very unclear whether your model fits for the individual athlete.

- Please provide an extra section for limitations.

Reviewer #3: This study investigates cardiac beat-by-beat measurements during maximal graded effort test; it can support other methods to consider the post-processing Heart Rate detections in promising tools for the study of cardiac measurements generated by portable devices with the aim of obtaining best performance in athletes. HRV was studied in function of the corresponding HR and it was confirmed that SDRR behaves as an exponential of the heart rate.

It seems very interesting, however, some points need to be better clarified. I think it is necessary, for example, in the introduction that a brief paragraph describing what CFR indices are and why it would be important to detect them.

The Akaike information criterion (AIC) and Bayesian Information criterion (BIC) can be briefly described in method or in appendix).

However, the main question is about the model that would describe the evolution of SDRR along the exercise (in the whole exercise duration time) by an exponential equation (in three different forms) in function of mean HR, power and workload. The parameter a and b in the equations are found by sliding windows along the exercise. Table 3 and Fig4 are refer only to model 1 (it may be better to change the title of the paragraph in “Study of individual decrease of HRV during effort in model 1”), this result must be emphasized.

The last part it is not clear in my opinion. How can Stationarity hypothesis in HR be assessed in window shorter than 30 sec (or 50 points)?

The fig.3 it is impressive, but it seems not to be necessary, or the significance must be better explained.

Specific comments

Abstract

Please rewrite the sentence: “It indicates that among athletes with better fitness, HRV has higher values at low heart rate and decreases faster when the heart rate increases during exercise”.

This is a key point. HRV has higher values at low heart rate it is well-known, it must be emphasized that HRV “decrease faster”

Introduction

How can measure the decrease of their variability in frequency domain?

Pag. 3

“such as the decrease of their variability measured both in time and frequency domain (8,10–15), the modification of the scaling properties of their linear correlations (16), or even the reduction of their sample entropy (17) and of their nonlinear correlations (18,19)”.

Too more information in two and half rows, please explain more in details and add even the more recent paper of [19] in PlosOne.

Pag 3 last row:

Add a definition and description cardiorespiratory fitness (CRF) indexes that are only mentioned at pag 4 :CRF indices (namely maximum VO2, maximum aerobic power, power at ventilator thresholds).

Pag 3 Row 15 replace work load with workload

Pag 3 i) Replace designed with defined

Please explain stationarity hypothesis in HR

Methods

Please, provide more information about sample.

For example: age, anthropometric measurements, explain why you can consider male and female together despite the well-known gender differences existing in physical exercises

calculated on adjacent windows of one minute (corresponding to each effort step during effort, see “Heart rate variability” subsection).

It is not clear and easy-readable: Please rewrite, describe the timing of protocol clearly, simply explain that the workload changes every minute. (see pag7)

Pag 7 “Detrending”: what are the differences among 23-24-25?

AIC and BIC can be briefly descried in method or in appendix

Pag 7 last rows

Because HRV is a result of parasympathetic and sympathetic neuronal activity, both being processes taking part in the regulation of the mean HR, we propose to analyze the decay of SDRR during exercise as a function of the mean HR calculated on the same time windows.

It is not clear because the second part (“we propose to analyze …” is a consequence of the first part “Because HRV is …)

Results

Pag 9 Explain better the first three rows

“The median number of RR distance recorded was 2726 beats (Inter Quartile Range [2433;3548]). An example of detrended RR estimated using the dynamical model presented in the method section is illustrated in Fig 2. This model produced an individual estimation of RR with a median R2 of 0.97 [IQR: 0.91 – 0.98].

Pag. 10 add to the first sentence something like: “for these reason in the following ……only model 1…”

In Fig3 Please add three panels with only “test”

In Caption of fig3 A comma is missing before “respectively”: in A, B and C, respectively

Pag 12 table please correct “correlation”

(HRV decay rate a and the HRV at HR=0 b) change “a, HRV decay rate and b, HRV decay rate at HR=0 “

HRV was studied in function of the corresponding HR:

Can you better explain the difference between �RR and �p ?

Please explain because it would be trivial to consider �RR and �p with the same number of points

In my opinion it is not clear to catch the differences between �RR and �p

Pag 13 athletes with higher aerobic capacity….. Add references

In the fig3, to what correspond the horizontal line below 50 appears in all the panels?

6. PLOS authors have the option to publish the peer review history of their article (what does this mean?). If published, this will include your full peer review and any attached files.

Reviewer #1: **Yes: **Yoshinari UEHARA

Reviewer #2: **Yes: **Thomas Gronwald

Reviewer #3: **Yes: **Giovanna Zimatore

---

## [Author Response · Author response to Decision Letter 0]

1 Jul 2022

Dear Professor Laurent Mourot, dear reviewers

Please find our revised version of our article «Decrease of heart rate variability during exercise: an index of cardiorespiratory fitness» and the associated detailed answer to the reviewers remarks.

We would like to thank the editorial board of PlosOne and the reviewers for the quality of the review. The remarks were pertinent and accurate, and helped us improving our manuscript. The detailed answers of each question can be found in the attached document

---

## [Decision Letter · Decision Letter 1]

19 Aug 2022

Decrease of heart rate variability during exercise: an index of cardiorespiratory fitness

PONE-D-22-11838R1

Dear Dr. Mongin,

We’re pleased to inform you that your manuscript has been judged scientifically suitable for publication and will be formally accepted for publication once it meets all outstanding technical requirements.

Kind regards,

Laurent Mourot

Section Editor

PLOS ONE

Additional Editor Comments (optional):

Reviewers' comments:

Reviewer's Responses to Questions

**Comments to the Author**

1. If the authors have adequately addressed your comments raised in a previous round of review and you feel that this manuscript is now acceptable for publication, you may indicate that here to bypass the “Comments to the Author” section, enter your conflict of interest statement in the “Confidential to Editor” section, and submit your "Accept" recommendation.

Reviewer #1: All comments have been addressed

Reviewer #2: All comments have been addressed

Reviewer #3: All comments have been addressed

2. Is the manuscript technically sound, and do the data support the conclusions?

Reviewer #1: Yes

Reviewer #2: Yes

Reviewer #3: Yes

3. Has the statistical analysis been performed appropriately and rigorously? 

Reviewer #1: Yes

Reviewer #2: Yes

Reviewer #3: Yes

4. Have the authors made all data underlying the findings in their manuscript fully available?

Reviewer #1: Yes

Reviewer #2: Yes

Reviewer #3: Yes

5. Is the manuscript presented in an intelligible fashion and written in standard English?

Reviewer #1: Yes

Reviewer #2: Yes

Reviewer #3: Yes

6. Review Comments to the Author

Reviewer #1: The authors have appropriately corrected all of the issues noted in the submitted paper. No further comments.

Reviewer #2: I thank the authors for adressing all of my my comments and questions! Changes made have improved the manuscript.

Reviewer #3: The research articles can be accepted for publication in PLOS ONE because it satisfied all the journal's criteria.

7. PLOS authors have the option to publish the peer review history of their article (what does this mean?). If published, this will include your full peer review and any attached files.

Reviewer #1: No

Reviewer #2: **Yes: **Thomas Gronwald

Reviewer #3: **Yes: **Giovanna Zimatore

---

## [Editor Report · Acceptance letter]

25 Aug 2022

PONE-D-22-11838R1 

Decrease of heart rate variability during exercise: an index of cardiorespiratory fitness 

Dear Dr. Mongin:

I'm pleased to inform you that your manuscript has been deemed suitable for publication in PLOS ONE. Congratulations! Your manuscript is now with our production department. 

Kind regards, 

on behalf of

Dr Laurent Mourot 

Section Editor

PLOS ONE